# Bright-Field Multiplex Immunohistochemistry Assay for Tumor Microenvironment Evaluation in Melanoma Tissues

**DOI:** 10.3390/cancers14153682

**Published:** 2022-07-28

**Authors:** Filippo Ugolini, Elisa Pasqualini, Sara Simi, Gianna Baroni, Daniela Massi

**Affiliations:** Section of Pathological Anatomy, Department of Health Sciences, University of Florence, 50139 Florence, Italy; filippo.ugolini@unifi.it (F.U.); elisa.pasqualini@unifi.it (E.P.); sara.simi@unifi.it (S.S.); gianna.baroni@unifi.it (G.B.)

**Keywords:** melanoma, bright-field multiplex immunohistochemistry, tumor microenvironment

## Abstract

**Simple Summary:**

Bright-field (BF) immunohistochemistry (IHC) remains the gold standard for histopathological evaluations. The development of new BF multiplex IHC could be very useful for the study and characterization of the tumor microenvironment (TME) in melanoma samples. We herein compared different BF IHC multiplex protocols for the study of TME in primary cutaneous melanoma tissues and offered the best optimized protocol for visualization and evaluation. These methodologies are studied to maximize the quality of staining considering the tissue characteristics under examination, maintaining a high level of standardization and reproducibility.

**Abstract:**

The tumor microenvironment (TME) plays a crucial role in melanoma development, progression and response to treatment. As many of the most relevant TME cell phenotypes are defined by the simultaneous detection of more than two markers, the bright-field (BF) multiplex immunohistochemistry (IHC) technique has been introduced for the quantitative assessment and evaluation of the relative spatial distances between immune cells and melanoma cells. In the current study, we aimed to validate BF multiplex IHC techniques in the Ventana Discovery Ultra Immunostainer to be applied to the evaluation of the TME in variably pigmented melanoma tissues. The BF multiplex IHC staining was performed using different combinations of six immune-cell markers—CD3, CD4, CD8, CD20, CD68 and CD163—and the melanoma cell marker SOX10. Our results show that the BF double IHC Yellow/Purple protocol guarantees the maximum contrast in all the cell populations tested and the combination SOX10 (Green), CD8 (Yellow) and CD163 (Purple) of the BF triple IHC protocol ensures the best contrast and discrimination between the three stained cell populations. Furthermore, the labeled cells were clearly distinct and easily identifiable using the image analysis software. Our standardized BF IHC multiplex protocols can be used to better assess the immune contexts of melanoma patients with potential applications to drive therapeutic decisions within clinical trials.

## 1. Introduction

Melanoma is one of the major causes of cancer-related death, and its incidence is increasing worldwide [1]. The immune system plays a crucial role in melanoma development and progression [2,3]. The evaluation of the TME can help to understand the mechanisms of response and resistance to immunotherapy, target therapy and guide clinical decisions for the individual patient.

Standard single chromogenic IHC has been applied to immune contextual evaluation in melanoma tissues as part of retrospective studies or exploratory analyses [4,5,6], mostly focusing on the prognostic and predictive impact of the density and distribution of immune cells, including tumor-infiltrating lymphocytes (TILs) and tumor-associated macrophages (TAMs) [7,8,9,10,11,12,13,14]. In the clinical workflow, inflamed (“hot”) melanomas show a high number of CD8+ T cells in the stromal compartment and within the tumor parenchyma, while non-inflamed “cold” melanomas are characterized by a scarce immune infiltrate, either as an immune desert or excluded pattern [15].

Although most pathologists are familiar with the semiquantitative assessment of immune cells by singleplex IHC, rough estimations must be annotated for consecutive tissue formalin-fixed and paraffin-embedded (FFPE) sections, making it difficult to compare cells to each other in limited tissues and impairing the interobserver reproducibility. The colocalization of cells of interest (at both the tumor center and invasive margin compartments) in the same section and evaluation of the relative distances between immune and tumor cells by spatial proximity analysis can provide more robust and clinically useful information [15,16,17].

Multiplex immunofluorescence (IF) staining allows the simultaneous or sequential detection of multiple markers in a single FFPE tissue section [18], and research discovery investigations have adopted different IF workflows, platforms and configurations. However, the use of IF in pigmented specimens in the routine clinical setting and confidence in their diagnostic use remain critical due to the high costs, long turnaround times, lack of standardized methods and technical challenges (including autofluorescence and crosstalk between different fluorophores with overlapping emission spectra) [19,20,21].

In contrast, bright-field (BF) multiplex IHC has been introduced to simultaneously visualize up to eight markers labeling different cell populations on the same section [22]. In recent years, in addition to stains already available (Silver, Purple, Blue, Yellow, RED and DAB), new colors such as Green and Teal have been introduced. However, there is currently limited knowledge on the best optimal combination and visual interpretation in variably pigmented melanoma tissues.

The purpose of this study was to validate BF multiplex IHC techniques in the Ventana Discovery Ultra Immunostainer that could allow the better standardization of TME assessment and increase confidence in the visual recognition of multiple markers in melanoma tissue specimens for clinical adoption and implementation.

## 2. Materials and Methods

### 2.1. Tissue Samples

FFPE primary cutaneous melanoma tissue sections (*n* = 20) 3 µm in thickness were obtained from paraffin blocks retrospectively selected from the Archive of the Section of Pathology, Department of Health Sciences, University of Florence, Florence, Italy.

### 2.2. Immunohistochemistry

TME characterization was performed by evaluating different BF multiplex IHC staining using different combinations of 6 immune-cell markers (CD3, CD4, CD8, CD20, CD68 and CD163) and a melanoma-specific cell marker (SOX10). Immunohistochemistry was performed on representative FFPE whole tumor sections 3 μm thick. We tested and validated the following BF multiplex IHC protocols: CD3/CD20, CD4/CD8, CD68/CD163 and CD163/CD8/SOX10. For all the BF multiplex protocols, sections were deparaffinized in EZ prep (#950-102; Ventana), and antigen retrieval was achieved by incubation with cell-conditioning solution 1 (#950-124; Ventana), a Tris ethylenediaminetetraacetic acid-based buffer (pH 8.2). Sections were incubated with the following primary antibodies: anti-CD3 (#790-4341, rabbit monoclonal, clone 2GV6 ready to use, Ventana Medical System, Tucson, AZ, USA), anti-CD4 (#790-4423, rabbit monoclonal, clone SP35, ready to use, Ventana Medical System, Tucson, AZ, USA), anti-CD8 (#790-4460, rabbit monoclonal, clone SP57, ready to use, Ventana Medical System, Tucson, AZ, USA), anti-CD20 (#760-2531, rabbit monoclonal, clone L26, ready to use, Ventana Medical Systems, Tucson, AZ, USA), anti-CD68 (#05721679001, mouse monoclonal, clone PGM1, ready to use, Diagnostic BioSystem, Pleasanton, CA, USA), anti-CD163 (#760-4437, mouse monoclonal, clone MRQ-26, ready to use, Ventana Medical Systems, Tucson, AZ, USA), and anti-SOX10 (#760-4968, rabbit monoclonal, clone SP267, ready to use, Ventana Medical Systems, Tucson, AZ, USA). Each denaturation step was performed by treating the slides with Ultra CC2 (#950-223, ready to use, Ventana Medical Systems, Tucson, AZ, USA) for 8 min. at 100 °C. The signal was developed with anti-mouse or anti-rabbit Alk Phos (AP) and anti-mouse or anti-rabbit HRP coupled with the following chromogens: DISC. PURPLE Kit (#760-229, ready to use, Ventana Medical Systems, Tucson, AZ, USA), DISC. GREEN HRP Kit (#760-271, ready to use, Ventana Medical Systems, Tucson, AZ, USA), DISC. YELLOW Kit (#760-239, ready to use, Ventana Medical Systems, Tucson, AZ, USA), Chromomap RED (#760-160, ready to use, Ventana Medical Systems, Tucson, AZ, USA) and Chromomap DAB (#760-159, ready to use, Ventana Medical Systems, Tucson, AZ, USA). The sections were counterstained with Hematoxylin II (#790-2208, ready to use, Ventana Medical Systems, Tucson, AZ, USA). For quantitative evaluations, we ran a singleplex BF multiplex IHC protocol for CD3, CD4, CD8, CD20, CD68 and CD163 for a subset of 10 melanoma samples randomly selected from the starting cohort. The signal was developed with anti-mouse or anti-rabbit Alk Phos (AP) coupled with Chromomap RED (#760-160, ready to use, Ventana Medical Systems, Tucson, AZ, USA). The sections were counterstained with Hematoxylin II (#790-2208, ready to use, Ventana Medical Systems, Tucson, AZ, USA).

### 2.3. Methodological Considerations

To optimize BF multiplex IHC procedures in the Ventana Discovery Ultra Immunostainer, it is strictly necessary to follow a rigorous methodology.

Indeed, the order of sequential staining is a crucial step in multiplexing procedures. To maximize the quality of the stain, it is always necessary to take into account (i) the antibody with the lowest antigenicity or that is more difficult to detect (weak staining), as in the singleplex method, must always be the first in the multiplex sequence; (ii) it is recommended to start the multiplex sequence with antibodies directed against nuclear antigens, then proceed with cytoplasmic ones and finally use those against the membrane; and (iii) the choice and the order of the chromogen combination influence the quality of the staining. If the antibodies of interest are expressed on the same cell populations, not all the chromogens available in BF are useful; only the Purple, Teal and Yellow ones, due to their translucent nature, can be combined for colocalization in the Ventana Discovery Ultra.

Moreover, another crucial step in BF multiplex IHC is the denaturation phase. We performed two different protocols of denaturation, using a combination of high temperature and reaction buffer (pH 8.2) or Ultra CC2 (pH 6), both with incubation for 8 min at 100 °C. Ultra CC2 as a denaturation buffer is more efficient than reaction buffer, and it guarantees cleaner and brighter colors by not reducing the previous chromogen staining at all. All the protocols that we propose in this short communication follow these recommendations.

### 2.4. Ethical Committee

The use of FFPE sections of human melanoma samples was approved by the Local Ethics Committee (13676_bio, protocol Id.21073) according to the Helsinki Declaration.

### 2.5. Image Analysis

Stained tissue sections were digitally scanned at ×400 magnification with the Aperio AT2 platform (Leica Biosystems, Wetzlar, Germany) into whole slide digital images (WSI). Each SVS-format file was imported into the HALO Link^®^ (Indica Labs, Albuquerque, NM, USA) image-management system. After the image annotations of the whole tumor areas of the melanoma samples were drawn, we performed image analysis for both singleplex and multiplex staining using the HALO Multiplex Immunohistochemistry analysis software version v3.1.1076.308 (Indica Labs, Albuquerque, NM, USA). The evaluations were based on cytonuclear features such as the stain intensity, size, and roundness for CD3-, CD4-, CD8,- CD20-, CD68- and CD163-positive cells. The software automatically excludes tissue gaps from analysis, and the settings were set up to include the full range of staining intensity (from weak to strong). The data are expressed as the cellular density (i.e., the number of positive cells divided by the annotation layer area in mm^2^).

### 2.6. Statistical Analysis

The data are reported descriptively, namely, as the means ± standard deviations (SDs) or medians and interquartile ranges (IQRs) for continuous variables, and numbers (percentages) for categorical data. The unpaired two-tailed Student’s *t* test was applied as appropriate. A *p*-value < 0.05 was considered significant for all the variables. All the data were analyzed using GraphPad Prism version 8.00 (La Jolla, CA, USA).

## 3. Results

### 3.1. Optimization of Double-Labeling Protocols

Starting from validated singleplex staining protocols routinely used, we first assessed whether these traditional BF IHC methods could be combined with each other; in particular, we combined six immune-cell-marker protocols for CD3, CD4, CD8, CD20, CD68 and CD163. These protocols allowed us to visualize two different immune-cell populations on the same section: CD3/CD20 (T and B lymphocytes), CD4/CD8 (T helper and cytotoxic T cells), and CD68/CD163 (M1 and M2 macrophages). We started to optimize the protocol from the CD3/CD20 staining (Figure 1).

Since these are antigens present on different cells on the membranes and with similar antigenicity, in this case, the order of staining did not affect the quality of the result (data not shown). Accordingly, to characterize the distribution of B cells and T cells in the TME of melanoma samples, we performed three different staining protocols using the following chromogen combinations: Red/DAB (Figure 1A), Yellow/Red (Figure 1B), and Yellow/Purple (Figure 1C). As shown in Figure 1, the BF multiplex IHC Yellow/Purple protocol guarantees the maximum contrast in comparison with the previous combinations tested (Red/DAB and Red/Yellow), allowing one to better distinguish the morphological and cyto-architectural tissue details. The excellent contrast and the distance of the colors in the visible spectrum make the Yellow/Purple combination the best-performing combination of chromogens for double labeling both for microscope evaluations and for image analysis. Moreover, as shown in Figure 2, we applied this protocol for CD4/CD8 (Figure 2B) and CD68/CD163 (Figure 2C) staining protocols. In order to validate the quality and the antigen conservation of the double-staining protocols shown in Figure 2B–D, we performed, on a subset of 10 melanoma samples, a singleplex Red staining for CD3, CD20, CD4, CD8, CD163 and CD68.

Unpaired-sample *t*-tests were run to determine if there were differences between CD3 singleplex/CD3 multiplex, CD20 singleplex/CD20 multiplex, CD4 singleplex/CD4 multiplex, CD8 singleplex/CD8 multiplex, CD163 singleplex/CD163 multiplex and CD68 singleplex/CD68 multiplex staining (Figure 2E). No differences were observed for CD3 singleplex (mean: 919.1 ± 565.9) and CD3 multiplex (mean: 983.0 ± 634.2), *p* = 0.081; for CD20 singleplex (mean: 206.5 ± 231.5) and CD20 multiplex (mean: 213.2 ± 237.8), *p* = 0.094; for CD4 singleplex (mean: 503.8 ± 484.5) and CD4 multiplex (mean: 577.4 ± 361.7), *p* = 0.070; for CD8 singleplex (mean: 333.1 ± 346.0) and CD8 multiplex (mean: 235.7 ± 189.7), *p* = 0.044; for CD163 singleplex (mean: 849.6 ± 560.1) and CD163 multiplex (mean: 1090.0 ± 550.4), *p* = 0.34; and for CD68 singleplex (mean: 391.8 ± 330.6) and CD68 multiplex (mean: 463.2 ± 198.4), *p* = 0.56. Quantitative evaluations confirm that there is no loss of antigenicity due to the multiple cycles of staining.

### 3.2. Optimization of Triple-Labeling Protocols

In order to study the relationship between immune cells and tumor cells, we developed and validated a triple-labeling BF IHC. We introduced the nuclear marker specific for melanoma cells SOX10 to our optimized double-labeling protocols. In Figure 3, we reported one of the possible combinations for triple-labeling BF IHC: SOX10/CD8/CD163. To achieve this goal, we introduced the green chromogen for the third stain. We tested three different chromogens’ combinations: SOX10 (Yellow), CD8 (Purple) and CD163 (Green) (Figure 3A); SOX10 (Purple), CD8 (Green) and CD163 (Yellow) (Figure 3B); and SOX10 (Green), CD8 (Yellow) and CD163 (Purple) (Figure 3C). Our results show that the combination SOX10 (Green), CD8 (Yellow) and CD163 (Purple) guarantees the best contrast and sharp discrimination between the three stained cell populations.

## 4. Discussion

With the growing relevance of the TME in the context of prognostic and predictive biomarkers’ discovery, implementations of novel and standardized methods for immune contextual evaluation are becoming crucial as a personalized approach to clinical decisions in melanoma patients. Singleplex IHC, while being cost-effective, practical and of widespread use in routine diagnosis, does not allow the accurate assessment of the density of and spatial relationships between different cell types. Today, the development of new multiplex IHC staining is increasingly in demand for clinical implementations that are in line with the recent transition towards fully digital diagnostic pathology [23,24,25,26].

IF and BF multiplex IHC staining coupled with image analysis allow a more consistent and in-depth view of the TME [27]. Hence, there is a need to put in place standardized protocols that are aimed at providing the best visual quality of markers that can be evaluated both by the operator and by the latest software generation.

From a technical point of view, unlike the singleplex methods, BF multiplex IHC requires a more in-depth study of each step of the protocol, because tissues undergo multiple cycles in staining procedures [28] that allow the visualization of more biomarkers of interest using a minimal number of slides, saving precious tissue samples.

Furthermore, in contrast to IF multiplex IHC, the development of assays involving BF multiple IHC chromogenic substrates presents some critical issues such as the visual contrast compatibility of chromogens and cellular localization of combined markers. IF multiplex IHC is undoubtedly a very powerful tool that allows a more complete and more in-depth characterization of the TME than the BF multiplex IHC [19,21,29]. IF multiplex IHC permits colocalizing more markers on the same cells than is technically impossible with BF multiplex IHC. In addition, the large number of dyes available and the possibility of “turning them on” and “turning them off” coupled with automated rapid imaging system LED technologies [23] make the IF multiplex IHC the leading staining technique in the research field [25,29]. For all these reasons, BF multiplex IHC has remained less widespread than IF.

On the other hand, BF IHC remains the gold standard for histological evaluations [29], and the development of BF multiplex IHC could be very useful for challenging specimens such as pigmented melanoma samples. The development of standardized BF multiplex IHC protocols could be more easily implemented in the diagnostic routine workflow since pathologists are more familiar with the evaluation of stained sections in BF than IF [30]. Another advantage of BF IHC is represented by ease of the acquisition of WSI systems [21]. Indeed, the acquisition of BF IHC WSI is performed through new completely automatic platforms that do not require specialized personnel thanks to the easy handling and storage of the BF IHC slides compared to IF slides.

To achieve this goal, we proposed three different CD3/CD20 double BF multiplex and three different triple BF multiplex SOX10/CD8/CD163 IHC protocols; our results showed that Yellow/Purple and Yellow/Purple/Green combinations yielded the best contrast, respectively. Indeed, the choice of complementary chromogen colors, opposite in the visible spectrum, such as purple and yellow, allows a reduced color overlap with a clearer visualization of the stained cell populations than the other chromogen combinations tested. Instead, the overlapping absorption spectra of Yellow/Red and the reduced contrast of Red/DAB could cause the misinterpretation of digital images and may affect the subsequent image analysis. Moreover, DAB is not recommended in pigmented melanoma tissues [31], and the use of this chromogen may increase the risk of improper interpretations, which result from the DAB chromogen intensifying the color of residual melanin pigment [32]. Of interest, highly pigmented cutaneous melanomas showed a worse prognosis in comparison with amelanotic tumors, suggesting that melanin can enhance melanoma progression possibly through an upregulation of multiple genes involved in glucose metabolism, angiogenesis and stress responses [33,34]. Consequently, an optimization of the characterization of the immunophenotype in highly pigmented melanomas remains of crucial relevance.

To obtain the best chromogens’ combination and visual interpretation in pigmented melanoma tissues, it may be useful to combine BF multiplex IHC with a bleaching method as we previously reported [35].

The use of complementary colors is essential to facilitate robust automated image analysis and processing by different AI-based image analysis software. Furthermore, we evaluated the antigen conservation of the tested protocols comparing the BF singleplex IHC with BF multiplex IHC, demonstrating that there was no loss of antigenicity due to the multiple cycles of staining.

All these considerations highlight the promising potential of the integration of BF multiplex IHC in clinical practice. The development and validation of these standardized techniques is crucial, particularly in consideration of the potential synergies with image analysis tools and artificial intelligence (AI) algorithms [26,36]. This could lead to the implementation of new AI tools and discovery of new prognostic factors that, in a future diagnostic routine, could assist pathologists in the diagnosis workflow.

In conclusion, we herein compared different BF IHC multiplex protocols for the study of the TME in primary cutaneous melanoma tissues and offer the best optimized protocol for visualization and evaluation. These methodologies are studied to maximize the quality of staining (choice of chromogens) considering the tissue characteristics under examination (variably pigmented specimens), maintaining a high level of standardization and reproducibility thanks to the complete automation of the process developed in the Ventana Discovery Ultra Immunostainer.

## Figures and Tables

**Figure 1 cancers-14-03682-f001:**
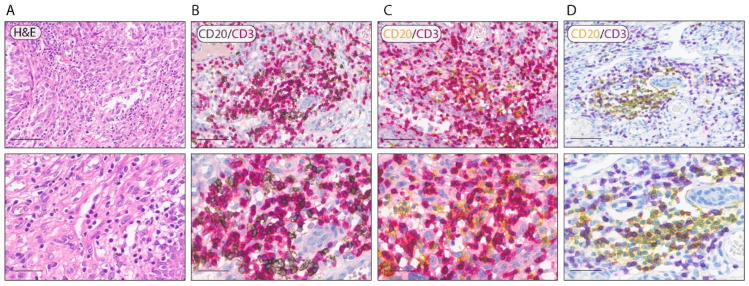
Representative double-labeling BF multiplex IHC of melanoma tissue. (**A**) Representative images of hematoxylin and eosin staining (**B**) Representative image of double CD20/CD3 staining with DAB (CD20) and Red (CD3) chromogens. (**C**) Representative image of double CD20/CD3 staining with Yellow (CD20) and Red (CD3) chromogens. (**D**) Representative image of double CD20/CD3 staining with Yellow (CD20) and Purple (CD3). Magnification: 200× and 400× (scale bar: 100 μm and 50 µm).

**Figure 2 cancers-14-03682-f002:**
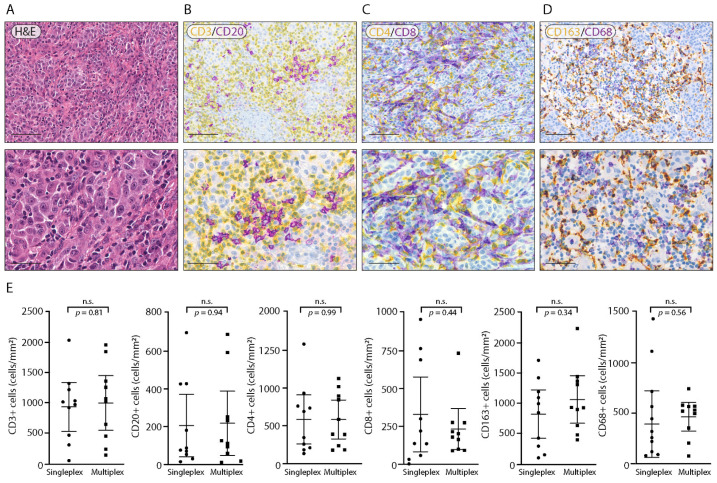
Representative double-labeling BF multiplex IHC of melanoma tissue and pooled data of CD3-, CD20-, CD4-, CD8-, CD163- and CD68-positive cell density comparisons between single and multiplex IHC. (**A**) Representative images of hematoxylin and eosin staining. (**B**) Representative image of double CD3/CD20 staining with Yellow (CD3) and Purple (CD20) chromogens. (**C**) Representative image of double CD4/CD8 staining with Yellow (CD4) and Purple (CD8) chromogens. (**D**) Representative image of double CD163/CD68 staining with Yellow (CD163) and Purple (CD68) chromogens. Magnification: 200× and 400× (scale bar: 100 µm and 50 µm). (**E**) Quantitative analysis of CD3-, CD20-, CD4-, CD8-, CD163- and CD68-positive cells in singleplex stained samples (*n* = 10) and multiplex stained samples (*n* = 10). Error bars represent 95% confidence intervals. All the statistical analyses were performed using unpaired two-tailed Student’s *t* tests. n.s.–not significant.

**Figure 3 cancers-14-03682-f003:**
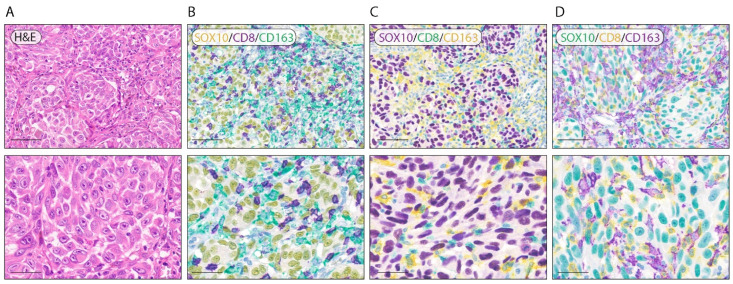
Representative triple-labeling BF multiplex IHC of melanoma tissue. (**A**) Representative images of hematoxylin and eosin staining. (**B**) Representative image of triple SOX10/CD8/CD163 labeling with Yellow (SOX10), Purple (CD8) and Green (CD163) chromogens. (**C**) Representative image of triple SOX10/CD8/CD163 staining with Purple (SOX10), Green (CD8) and Yellow (CD163) chromogens. (**D**) Representative image of triple SOX10/CD8/CD163 staining with Green (SOX10), Yellow (CD8) and Purple (CD163) chromogens. Counterstain was performed with hematoxylin. Magnification: 200× and 400× (scale bar: 100 µm and 50 µm).

## Data Availability

The data presented in this study are available on request from then corresponding author.

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
