# Peer review of "Bright-Field Multiplex Immunohistochemistry Assay for Tumor Microenvironment Evaluation in Melanoma Tissues"

_cancers, 2022, doi:10.3390/cancers14153682_

Round 1
Reviewer 1 Report
This manuscript describes complementary chromogens colors for bright-field IHC staining that yield the best contrast to detect tumor cells and T- and B- cells, a feature critical to evaluate the relationship between immune cells and tumor cells in the microenvironment. This information is critical to evaluate melanoma patients undergoing immune- or targeted therapy.
Author Response
Point 1: This manuscript describes complementary chromogens colors for bright-field IHC staining that yield the best contrast to detect tumor cells and T- and B- cells, a feature critical to evaluate the relationship between immune cells and tumor cells in the microenvironment. This information is critical to evaluate melanoma patients undergoing immune- or targeted therapy.
Response 1: We thank the reviewer for his/her positive evaluation.
Reviewer 2 Report
Authors present here a novel methodology for the visualization of immune cells in melanoma. As the infiltration of tumors by activated immune cells strongly determines the patient´s outcome and response to therapeutic intervention, the development of novel techniques for visualization and quantification of tumor cells subsets and microenvirnomental cells is highly important. However, some minor points needs to be addressed:
1.) authors mentioned the parallel staining of up to 8 markers but images mostly show only two or three markers. I understand that overlaying more than two/three markers would be accompanined by a reduced detection/resolution of cellular subsets however, can you provide a staining with more than three markers?
2.) You mentioned that images were loaded in the HALO software. As far as I know this also allows a quantification of cellular subsets. Visulaization on one hand is important, however the automated quantification of cellular subsets on the other hand serves as a prognostic tool. Why authors did not perform a quantification of cellular subsets?
3.) Expression of PD-L1/PD1 -although this is not that frequently observed in melanoma- serves as a prognostic marker. Did authors investigate levels of PD-L1 or PD-L2 in these samples?
Author Response
Point 1: Authors mentioned the parallel staining of up to 8 markers but images mostly show only two or three markers. I understand that overlaying more than two/three markers would be accompanined by a reduced detection/resolution of cellular subsets however, can you provide a staining with more than three markers.
Response 1: We thank the reviewer for his/her comment. We are currently trying to validate a 4-color BF multiplex IHC. To date we are not yet ready to show the data of the 4-color stain due to color overlap issues for image analysis. Now we are testing many different 4-colors combination in order to set up the best conditions for the quantitative analysis.
Point 2: You mentioned that images were loaded in the HALO software. As far as I know this also allows a quantification of cellular subsets. Visulaization on one hand is important, however the automated quantification of cellular subsets on the other hand serves as a prognostic tool. Why authors did not perform a quantification of cellular subsets?
Response 2: We thank the Reviewer for her/his comment. We run a control of singleplex stain for CD3, CD4, CD8, CD20, CD68 and CD163 in a subset of 10 samples randomly chosen. We quantified by HALO digital Multiplex immunohistochemistry analysis software (Indica Labs) cellular densities both for singlexplex and multiplex stains. We demonstrated that there are no significant differences in the cellular quantification between singleplex and multiplex staining procedures.
Point 3: Expression of PD-L1/PD1 -although this is not that frequently observed in melanoma- serves as a prognostic marker. Did authors investigate levels of PD-L1 or PD-L2 in these samples?
Response 3: We thank the reviewer for her/his observation. We have already investigated the expression of PD1 and PDL1 in a larger cohort of primary cutaneous melanoma (see De Logu et 2021 https://doi.org/10.3390/cells10020422). The purpose of this study is to validate multiplex BF techniques for the characterization of the tumor microenvironment in melanoma tissues, to date, for technical reasons, the colocalization of multiple markers using the BF multiplex IHC is not accepted. For these reasons, we have not developed multiplex protocols with PD-L1, PD-L2, PD1 and immune cells markers. In addition, the setup of the image analysis would be very challenging, since the overlap of the colors in BF on the same cell population is very hard to detect.
Reviewer 3 Report
These are predominantly descriptive studies to evaluate melanoma tumor and its microenvironment using IHC. I do not have a major critique as relates to methodology and direct data analysis.
My concern relates to the limited scope of the study. Presentation of the immunocytochemistry only without any effort to correlate the finding with changes in tumor behavior or other clinicopathologic parameters.
Therefore, conceptual advancement will be limited. Even in this limited, rather technical in nature study, that authors could correlate their stain with parameters of tumor progression.
Also with melanin pigmentation, mentioned by authors, taking into consideration that melanogenesis and melanin can affect melanoma behavior (Frontiers in Oncology 2022;12. DOI: 10.3389/fonc.2022.842496; Arch Biochem Biophys 563:79-93, 2014)
Author Response
Point 1: These are predominantly descriptive studies to evaluate melanoma tumor and its microenvironment using IHC. I do not have a major critique as relates to methodology and direct data analysis.
My concern relates to the limited scope of the study. Presentation of the immunocytochemistry only without any effort to correlate the finding with changes in tumor behavior or other clinicopathologic parameters.
Therefore, conceptual advancement will be limited. Even in this limited, rather technical in nature study, that authors could correlate their stain with parameters of tumor progression.
Also with melanin pigmentation, mentioned by authors, taking into consideration that melanogenesis and melanin can affect melanoma behavior (Frontiers in Oncology 2022;12. DOI: 10.3389/fonc.2022.842496; Arch Biochem Biophys 563:79-93, 2014).
Response 1: We thank the reviewer for her/his observation. We have already investigated the correlation of the markers included in this study with clinicopathologic parameters in a larger cohort of primary cutaneous melanoma (see De Logu et 2021 https://doi.org/10.3390/cells10020422). The aim of this study is to validate BF multiplex IHC protocols couple with advanced image analysis to give the possibility to explore new histopathological parameters such as relative cells distances that could represent new important histopathological parameters in melanoma. Thus, the development of these protocols represents an important starting point that could be used later in larger cohorts of patients to correlate the immunohistochemical results with clinicopathological parameters.
Reviewer 4 Report
In the article "Bright-field assay for multiplex immunohistochemistry tumor microenvironment evaluation in melanoma tissues" Ugolini et al., have developed bright field triple IHC protocol ensures the best contrast and discrimination between the three stained cell populations. The work is of great importance in the disease pathology and companion diagnostic analyses. However, I have several reservations about the publication with the current data. The quantification methodology in the manuscript is completely missing. Authors can address the following concerns before it could be accepted.
1. Authors have stained dual colors and tri colors, but no quantification was provided whatsoever in the entire manuscript, the staining must be quantified that is the goal of the histology. The colors need to be quantified individually and in complex. For example, in Figure 1A, the authors need to quantify CD20 and CD3 in each of the individual analysis from the sample set and validate statistically, how similar are different they are or how different.
2. Authors also need to perform similar quantification for other staining in Figure 2 and Figure 3.
3. The most important aspect of co-staining is we tend to see the decrease in each stain due to color bleaching or discoloration by another color. If authors can run a control of single color on adjacent slide and quantify them, it would be a better measure to see the staining efficiency.
4. The quantifications need to include n values and individual datapoints.
5. If authors can also provide H&E image with that would demonstrate the tissue architecture.
Author Response
Point 1: Authors have stained dual colors and tri colors, but no quantification was provided whatsoever in the entire manuscript, the staining must be quantified that is the goal of the histology. The colors need to be quantified individually and in complex. For example, in Figure 1A, the authors need to quantify CD20 and CD3 in each of the individual analysis from the sample set and validate statistically, how similar are different they are or how different.
Point 2: Authors also need to perform similar quantification for other staining in Figure 2 and Figure 3. The most important aspect of co-staining is we tend to see the decrease in each stain due to color bleaching or discoloration by another color. If authors can run a control of single color on adjacent slide and quantify them, it would be a better measure to see the staining efficiency. The quantifications need to include n values and individual datapoints.
Response 1 & 2: We thank the Reviewer for her/his comment. As requested, we run a control of singleplex stain for CD3, CD4, CD8, CD20, CD68 and CD163 in a subset of 10 samples randomly chosen. Then we quantified by HALO digital Multiplex immunohistochemistry analysis software (Indica Labs) cellular densities both for singlexplex and multiplex stains. We demonstrated that there are no significant statistical differences in the cellular quantification between singleplex and multiplex. Data include n values and single data point as requested.
Point 3: If authors can also provide H&E image with that would demonstrate the tissue architecture.
Response 3: Images of hematoxylin and eosin have been added in figures 1, 2 and 3.
Round 2
Reviewer 3 Report
The authors failed to address the critique of the reviewer #1.
Please discuss melanin pigmentation with proper referral as listed in the original critique
Author Response
We thank the reviewer for his/her comments. We added a paragraph in the discussion section. The two references: Frontiers in Oncology 2022;12. DOI: 10.3389/fonc.2022.842496; Arch Biochem Biophys 563:79-93, 2014 were cited in the body text and added in references list.
Reviewer 4 Report
The authors have addressed the appropriate concerns.
Author Response
We thank the reviewer for his/her positive evaluation.